# Phage Display-Based Nanotechnology Applications in Cancer Immunotherapy

**DOI:** 10.3390/molecules25040843

**Published:** 2020-02-14

**Authors:** Martina Goracci, Ymera Pignochino, Serena Marchiò

**Affiliations:** 1Department of Oncology, University of Torino, 10060 Candiolo, Italy; 2Candiolo Cancer Institute, FPO–IRCCS, 10060 Candiolo, Italy

**Keywords:** cancer, immunotherapy, phage display, vaccine, peptide, nanocarrier

## Abstract

Phage display is a nanotechnology with limitless potential, first developed in 1985 and still awaiting to reach its peak. Awarded in 2018 with the Nobel Prize for Chemistry, the method allows the isolation of high-affinity ligands for diverse substrates, ranging from recombinant proteins to cells, organs, even whole organisms. Personalized therapeutic approaches, particularly in oncology, depend on the identification of new, unique, and functional targets that phage display, through its various declinations, can certainly provide. A fast-evolving branch in cancer research, immunotherapy is now experiencing a second youth after being overlooked for years; indeed, many reports support the concept of immunotherapy as the only non-surgical cure for cancer, at least in some settings. In this review, we describe literature reports on the application of peptide phage display to cancer immunotherapy. In particular, we discuss three main outcomes of this procedure: (i) phage display-derived peptides that mimic cancer antigens (mimotopes) and (ii) antigen-carrying phage particles, both as prophylactic and/or therapeutic vaccines, and (iii) phage display-derived peptides as small-molecule effectors of immune cell functions. Preclinical studies demonstrate the efficacy and vast potential of these nanosized tools, and their clinical application is on the way.

## 1. Introduction

The immune system changes and adapts during the progressive steps of tumorigenesis to recruit and activate all possible mechanisms of protection. The initial response provides a first line of defense that keeps the neoformation in check. During this phase, the innate immune system recognizes cancer cell neoantigens (often called tumor-associated antigens (TAAs)) the same way it detects the presence of foreign organisms, e.g., pathogenic bacteria or viruses. Upon interaction with circulating TAAs, specific clones of B lymphocytes are amplified to produce high titers of anti-TAA antibodies (Abs). Antigen-presenting cells (APCs) bind, process, and expose the TAA on the major histocompatibility complex (MHC). Naïve T lymphocytes are recruited and primed upon interaction of T-cell receptors (TCRs) and co-receptors (expressed on T lymphocytes) with TAA-MHC complexes and co-ligands (expressed on APCs), respectively [1,2]. When the TAA is presented on MHC I, the complex is recognized by CD8+ cells that differentiate in cytotoxic T lymphocytes (CTLs) and kill cancer cells [3] causing the release of additional TAAs and amplification of the immune response [4]. When the TAA is loaded on MHC II, CD4+ T lymphocytes are engaged to differentiate in T helper (Th) lymphocytes that support the functions of B lymphocytes. T-cell co-receptors (the so-called “immune checkpoints”) can have a stimulatory or inhibitory effect. CD28 was the first co-stimulatory molecule to be identified, and the interaction between CD28 and B7.1/B7.2 is required for full activation of naïve T lymphocytes. In contrast, CTL-associated protein 4 (CTLA-4) and programmed cell death 1 (PD-1), when interacting with their ligands PDL-1/PDL-2 and B7.1/B7.2, respectively, restrain the activation of T lymphocytes and are thus considered co-inhibitory molecules [5].

While efficiently controlling tumor growth, immune responses also introduce a strong selective pressure that, with time, impacts on cancer cells, making them able to elude the immune surveillance and keep growing via genomic instability, anti-apoptotic pathways, angiogenesis, and eventually, metastatic spreading [6,7]. In addition, self-recognition mechanisms—whose physiological function is to prevent the immune system from attacking the host—may as well reinforce the escaping potential of cancer cells. Cancer immunotherapy is based on inducing a person’s immune system to attack tumor cells [8,9]. It involves a wide variety of therapeutic modalities such as Abs, vaccines, cytokines, and cell therapies that either stimulate antitumor responses (by activating effector cells) or impair suppressor mechanisms (by blocking the immune checkpoints). In a nutshell, immunotherapy boosts the innate power of our organism to combat cancer [10].

Conventional approaches for the therapeutic induction of anti-cancer immune responses follow three general concepts. First, anti-TAA Abs may be administered to patients, a procedure known as passive immunization. Since the invention of the hybridoma technology by Kohler and Milstein in 1975 [11], enormous efforts have been made on the characterization of monoclonal Abs as tumor-targeting agents. Several Abs have been developed against extracellular, transmembrane, or intracellular proteins to nullify the pathological mechanisms implemented by tumor cells, many of which have received Federal Drug Administration (FDA) approval [12,13,14,15,16]. These applications, however, suffer of some drawbacks that range from engineering human/humanized Abs to the need of repeated administrations to obtain effective serum levels of the Ab [17]. Ideally, a continuous and endogenous production of anti-TAA Abs would best serve the aim of achieving a therapeutic effect with the potential to eradicate the disease and prevent its recurrence for the rest of a patient’s life. This second concept, known as active vaccination, was historically developed to combat pathogenic organisms but has been recently pursued also in oncology. In this context, TAAs or TAA-mimicking molecules (mimotopes), soluble or conjugated to different nanocarriers, have been tested as cancer vaccines [18,19,20,21,22]. Finally, the third concept is a straightforward approach that has been defined “meddle with meddlers” [23], as in directly targeting the immune checkpoint inhibitors.

These three concepts have been applied to cancer treatment by means of different tools (e.g., peptides, Abs, aptamers, chemical molecules) developed with conventional techniques as discussed elsewhere [24,25,26]. Here, we focus on peptide phage display, performed with many variations from the original protocol, often in combination with advanced computational modelling. Phage display is a high-throughput proteomic method based on viruses that infect bacteria, the bacteriophages (in short, phages). These viruses vary widely in size (from 28 × 28 nm of phage Qβ to 900 × 7 nm of phage M13) and shape (icosahedral, with or without tail, filamentous), yet sharing important common features: they (i) naturally and quickly propagate in their bacterial host and are therefore suitable for molecular manipulations, (ii) are capable of self-assembly to provide ready-to-use nanosized tools, (iii) are easily engineered to expose one or more copies of a peptide fused to their capsid proteins. Starting from a library of millions of unique motifs, successive rounds of affinity selection are applied to isolate target-specific peptides. From this point on, the applications are manifold. The general uses and characteristics of phages as peptide-directed vectors for targeted nanomedicine are extensively discussed in the literature, for example in the comprehensive reviews by Yao [27] and Hess [3] and colleagues. The aim of the present review is to specifically address the applications of phage display nanotechnology in the field of immuno-oncology and to provide a broad overview of the different approaches explored in animal models, and in some cases, also in early-phase clinical trials.

## 2. Mimotopes of Tumor-Associated Antigens

Due to their specific and often very high expression, TAAs are attractive targets for cancer therapy. Ways to trigger the production of anti-TAA Abs include administration of either full-length TAAs, their antigenic parts only, or TAA mimotopes (e.g., anti-idiotype Abs or peptides recognized by the anti-TAA Ab). There is a vast literature on phage display panning of either monoclonal or polyclonal Abs to identify TAA mimotopes, in some cases even without knowledge of the corresponding natural epitope (see, e.g., [28]).

### 2.1. Mimotopes of CD20

CD20 is a 33–35-kDa non-glycosylated phosphoprotein expressed on B lymphocytes from the early pre-B to the late-B stage. Its expression is lost after differentiation of B lymphocytes into plasma cells, but the majority of human B-lineage malignancies re-express CD20, making it a likely targetable TAA. Panning the anti-CD20 Ab rituximab with phage-displayed peptide libraries has provided several mimotopes that could be developed as cancer vaccines. Among these, the linear 12-mer R10-L (ITPWPHWLERSS), despite having no homology to CD20, was shown to interact with rituximab and inhibit its binding to CD20. Anti-CD20 Abs were present in sera from BALB/c mice immunized with R10-L, as confirmed by binding assays on the CD20+ human B-lymphoid cell line Raji [29]. A mimotope identified in an independent panning, QDKLTQWPKWLE, revealed a 6-aa motif with >80% identity to R10-L and was shown to inhibit rituximab to Raji cells. This peptide was coupled to keyhole limpet hemocyanin (KLH) or tetanus toxoid (TT) and evaluated in experimental vaccinations of BALB/c mice. Sera raised against either QDKLTQWPKWLE-KLH or QDKLTQWPKWLE-TT recognized cell-surface CD20 and specifically killed Raji cells in complement-dependent cytotoxicity assays [30]. In a successive work from the same group, alignment of several phage-selected 7- and 12-mers confirmed the same 6-aa motif (WPxWLE) that the authors assigned to a reverse-oriented portion (_161_WPKWLE_156_) of acid sphingomyelinase-like phosphodiesterase 3b (ASMLPD). Corresponding synthetic peptides inhibited both rituximab binding and rituximab-induced, complement-dependent cytotoxicity in a specific and dose-dependent manner. Sera from BALB/c mice immunized with WPxWLE peptides reacted with both Raji cells and the human Burkitt’s lymphoma cell line Daudi (both CD20+) but not with the human T cell leukemia cell line CEM (CD20-). Further selections with phage-displayed libraries exposing cyclic 7-mers allowed to identify peptides that share the (A/S)NPS motif mapping to the extracellular portion of human CD20 (_170_ANPS_173_). Sera raised against these peptides were cytotoxic for Raji and Daudi cells in the presence of complement [31]. A panel of eleven variants, all including the rituximab-specific antigenic motif but with different flanking residues, were used for epitope mapping. The panel was narrowed to R13-C (WAANPS, identical to CD20 epitope) and R15-C (PYANPSL, flanking residues different from the natural epitope), which exhibited similar reactivity to rituximab. In vivo, R13-C elicited anti-CD20 Abs more consistently than R15-C (5 out of 5 versus 2 out of 5 mice), demonstrating the contribution of epitope-surrounding residues to Ab specificity [32]. Previously identified as mapping to ASMLPD, R5-L (WPKWLE) was successively characterized as an antigenic mimic of raft-associated CD20, mapping to a different epitope from, and showing no cross-inhibition with R15-C (PYANPSL). Computer modelling of possible contacts between R5-L/R15-C and rituximab indicated that these two motifs have similar—yet not identical—tertiary interactions with the Ab, and that they share some contact points within the rituximab/antigen binding site [33]. Abs against R5-L were produced by hybridoma technology, and clones FE-718 and FE-341 were chosen for further studies. Both Abs dose-dependently, and to a similar extent, inhibited rituximab binding to R5-L in a competitive binding assay. However, neither FE-718 nor FE-341 were able to detect cell-surface CD20. Epitope mapping of these Abs by phage display provided the consensus sequence WPxxL, similar to R5-L and recognized by sera from R5-L-immunized BALB/c mice. So, these CD20 mimotopes and their corresponding Abs exhibit both common and divergent features that can be exploited to design therapeutic tools. Further, these studies demonstrate that a 6-aa peptide motif (much shorter than the whole extracellular portion of CD20) can induce various immune responses [34] and that phage display is a powerful nanotechnology to get insight into the molecular basis of epitope spreading.

### 2.2. Mimotopes of the Epidermal Growth Factor Receptor

The epidermal growth factor receptor (EGFR, also called HER or ErbB) is a member of the EGFR family found overexpressed and/or mutated in many solid tumors, e.g., lung, neck, breast, kidney, and colon. This feature makes EGFR a great target for cancer treatment. Several anti-EGFR Abs have been developed, among which cetuximab, panitumumab, nimotuzumab, 12H23, Ch806, and ICR-6214. Although in early-stage clinical studies, prolonged treatment with cetuximab or matuzumab was generally well-tolerated with skin reactions such as an acne-like rash being the most common side effect, once phase II was reached, some anti-EGFR Abs showed disappointing results and scheduled trials were cancelled (matuzumab being one of them). A way to circumvent this drawback is to develop alternative approaches by exploiting the mimotope strategy. A number of mimotopes recognized by those Abs have been discovered with the phage display technique and mimotope-based vaccines have been consequently designed. Four peptide mimics of the epitope recognized by cetuximab were isolated from a cyclic 10-mer library, although none shared sequence similarities to either EGFR or other family members. Based on the specific binding to cetuximab, two of them (CQFDLSTRRLKC and CQYNLSSRALKC) were chosen for in vivo immunization studies after coupling with the immunogenic carrier KLH. Both mimotopes elicited Abs capable of triggering (i) cellular and complement-dependent cytotoxic effects against EGFR-expressing cells, (ii) EGFR internalization, and (iii) dose-dependent inhibition of proliferation in the EGFR-overexpressing human squamous carcinoma cell line A431 [35]. In another work, different phage-displayed peptide libraries (liner 7-mer, linear 12-mer, and cyclic 7-mer) were panned on cetuximab and matuzumab single-chain Fv Ab fragments (scFv) presented by *Escherichia coli* cells, with a procedure known as delayed infectivity panning [36]. The peptide motifs KTL and YPLG were retrieved as antigenic for both cetuximab and matuzumab, despite the non-overlapping EGFR epitopes previously reported for these Abs. In experimental immunizations of mice, synthetic KTL- and YPLG-containing peptides elicited Abs that recognize cell-surface EGFR and compete with natural ligands for receptor binding and activation, with an efficacy similar to cetuximab and matuzumab. Competition experiments demonstrated that the epitopes recognized by the anti-KTL and anti-YPLG Abs are close to, or overlap with, the binding sites of cetuximab, matuzumab, or the natural ligands transforming growth factor α (TGF-α) and EGF [37].

Mimotopes for other anti-EGFR Abs have also been selected, although with debatable results. Two potential peptide mimics of EGFR (WHTEILKSYPHE and LPAFFVTNQTQD) were identified by phage display panning of Abs 12H23 and Ch806. Mice immunized with WHTEILKSYPHE-KLH or LPAFFVTNQTQD-KLH developed high-titer Abs that recognized both EGFR and its variant EGFRvIII overexpressed on cancer cells. The mimotopes of this study were capable of inducing Ab-dependent but not complement-dependent cytotoxicity [38]. Another EGFR mimotope (QHYNIVNTQSRV) was identified by phage selection on Ab IRC-62 and validated by enzyme-linked immunosorbent assay (ELISA). A synthetic version of this peptide, conjugated to bovine serum albumin (BSA), was administered to rabbits in a vaccination experiment. The induced Abs bound to purified EGFR and inhibited the growth of A431 cells while having no effect on the EGFR-negative human melanoma cell line MDA-MB-453 [39]. In a similar work, epitope mapping of the same Ab, ICR-62, led to the isolation of a mimotope-displaying phage that was used in vaccination experiments, although with poor results, i.e., low titers of induced anti-EGFR Abs and neither prophylactic nor therapeutic efficacy in an in vivo model of Lewis lung carcinoma [40]. Finally, two other mimotopes, P19 (DTDWVRMRDSAR) and P26 (VPGWSQAFMALA), were selected by panning the anti-EGFR Ab panitumumab. Molecular modelling showed that, despite no sequence homology with EGFR, both P19 and P26 recapitulate the conformational structure of EGFR surface binding to panitumumab. Coupled with heat-shock cognate protein 70 (Hsc70) as an immunogenic carrier, both P19 and P26 stimulated the production of specific Abs in mice. The conjugates induced both Ab- and complement-dependent cytotoxicity and inhibited the proliferation of A431 cells. In addition, treatment with Hsc70-P19 and Hsc70-P26 reduced tumor growth in severe combined immunodeficient (SCID) mice bearing xenografts of the human lung cancer cell line A549 [41].

### 2.3. Mimotopes of HER2

HER2 (ErbB-2 or Her-2/neu), the product of the c-erbB-2 protooncogene, is a transmembrane, ligand-less tyrosine kinase receptor found to be overexpressed in several tumors, among which ovarian, prostate, and breast, whereas it is rarely expressed in normal adult tissues. Transtuzumab (Herceptin^®^), a humanized version of mouse anti-ErbB-2 clone 4D5, is clinically used to treat metastatic breast tumors that overexpress HER2. However, chimeric or humanized Abs are still recognized as foreign proteins and can trigger severe side effects due to hypersensitivity reactions, thus requiring a careful management of treatments. With this premise, a mimotope-based vaccination strategy may represent an option to elicit endogenous antitumor Abs while inducing a long-lasting immunologic memory. In an initial study, candidate mimotopes isolated by panning trastuzumab with a cyclic 10-mer phage-displayed library were validated as HER2 mimics in both direct and competitive ELISA. Immunization of BALB/c mice was performed with one of the selected peptides (CQWMAPQWGPDC) conjugated to TT, resulting in the production of Abs that recognized HER2 both in Western blot and on the surface of the HER2+ human breast cancer cell line SK-BR-3. Similar to trastuzumab, these Abs also induced HER2 internalization [42]. A different approach was applied in a more recent study, where an adeno-associated virus (AAV) library was panned on trastuzumab to identify HER2 mimotopes. The aim of this work was to obtain a vector suitable for immunization trials without further modification. Seven candidate peptides of consensus sequence WxxGxAxGS were tested in vivo and two were classified as optimal candidates based on (i) capacity to induce an Ab response, (ii) specificity of the induced Abs, and (iii) capacity to inhibit proliferation when applied to HER2-overexpressing cancer cells. Prophylactic vaccination of mice prior to subcutaneous grafting with HER2+ syngeneic D2F2/E2 cancer cells resulted in delayed tumor growth [43].

Phage display has also been applied to anti-HER2 Abs L26, N12, and L288. Of the selected motifs, peptide 52 (ALVRYKDPLFVWGFL), mapping to a portion of HER2 close to the natural epitope, competed for the binding of HER2 to L26 [44]. In a further study, epitope mapping of the anti-HER2 Ab SER4 provided another potential mimotope (INNEYVESPLYM) that shares homology with the region _87_AHNQVRQVPLQR_98_ in the extracellular domain of HER2 [45]. However, to our knowledge, there has been no follow up to these potentially promising studies.

### 2.4. Miscellaneous Mimotopes

Another line of research that started early in the era of phage display is the search for mimotopes of prostate membrane specific antigen (PSMA, also called prostate specific antigen, PSA). Three motifs (VDPGKYNKY, EGPAKGFKL, and GCYEAPSKAAKC) were isolated from panning the anti-PSA Ab 4G5. Sequence analysis demonstrated homology with a liner epitope within the extracellular portion of PSA (aa 719-725) [46]. In a more recent study, ten clones were isolated from a peptide phage display screening on a commercial anti-PSA Ab, four of which (RRSHPCRTCTTHTP, HRKTTCTRCPATSP, HRRGECRACPLLPA, and RRPAHCHHCPRNP) were further characterized. Their immunogenicity was determined by vaccination of BALB/c mice, with each peptide inducing Abs with minimal cross-reactivity to each other. In vitro, Abs specific to one peptide were capable of recognizing full-length PSA by Western blot, further confirming the mimotope properties [47].

In addition to PSA, other tumor markers have been investigated, such as the carcinoembryonic antigen (CEA), a 180-kDa glycoprotein composed by ~50% carbohydrates and frequently expressed in adenocarcinomas of various tissues of origin. Two phage-displayed peptide libraries (cyclic 10-mer and linear 9-mer) were screened on the anti-CEA Ab Col-1. The motif showing the highest specificity (DRGGLWKTP) was synthesized as an octameric multiple antigenic mimotope (MAM) of formula (DRGGLWKTP-GG)_4_-(KKGGC)_2_-dithioacetylhexandiamin. BALB/c mice immunized with this MAM mounted a specific anti-CEA humoral response (prevalently IgM and IgG_2_, with limited amounts of IgG_1_ and IgG_2a_). In vitro, these Abs triggered both Ab- and complement-dependent cytotoxicity against the CEA+ human colorectal cancer cell line HT29, but not against the CEA- cell line SW480. In vivo, immunization with the MAM led to delayed tumor growth in mice challenged with CEA-overexpressing syngeneic Meth-A fibrosarcoma cells [17].

Finally, phage display has also been applied to identify peptides specific for bevacizumab (Avastin^®^), a humanized anti-vascular endothelial growth factor (VEGF) monoclonal Ab approved for anti-angiogenic treatment of cancer. One of the selected peptide motifs (12P, DHTLYTPYHTHP) showed selective binding to bevacizumab and was further characterized. A synthetic version of the peptide was produced and conjugated to KLH to be used as a potential vaccine (KLH-12P). Mice injected with KLH-12P mounted a substantial humoral response, and sera from immunized mice inhibited VEGF binding to its receptor, as well as cell proliferation and migration [48].

### 2.5. Mimotopes of Unknown Antigens

A number of Abs recognizing the surface of gastric cancer cells have been identified, whose corresponding antigens remain uncharacterized. In this setting, the search for Ab-binding mimotopes is expected to provide insight into the nature of the TAA itself, in light of perspective clinical applications. A research group based in China generated a panel of monoclonal Abs against gastric cancer, designated as MG series. In one study, a phage-displayed 7-mer library was panned on MGb1 Ab providing potential mimotopes with the common motifs HxQ and LxS. A synthetic version of these peptides showed weak competition (20%) for the binding of MGb1 Ab to the human gastric cancer cell line KATO III. These results do not appear particularly promising, especially in the absence of in vivo evidence [49]. In another study from the same group, libraries of phages displaying 9-mer peptides in linear or cyclic form were used for epitope mapping of MG7 Ab. Phages exposing selected peptides proved to be specific both in dot blot and ELISA, and convincingly (>70%) competed for binding to KATO III cells. Immunization of BALB/c mice produced antisera, which, however, failed to react with gastric cancer cells, except for sera against GC8 (NAIYARNAQ), GC9 (TCHLRVYAQ), and GC28 (SWAPVYARN) [50]. Additional peptides selected by phage display on MG7 Ab were evaluated in two therapeutic strategies. Firstly, a single MG7 mimotope (KPHVHTK) [51] was associated to adjuvant CpG oligodeoxynucleotides in nanoemulsion. Immunization with these nanoemulsion induced interferon γ (IFN-γ) production in splenocytes—as determined by Enzyme-Linked ImmunoSPOT (ELISPOT)—and triggered a humoral response against MG7 antigen [52]. Secondly, four MG7 mimotopes (the previously cited KPHVHTK, plus KPHLHFH, KPHSHLH, and SWAPVYARAN) were individually conjugated to Hsp70 and combined in a multi-epitope vaccine. In vivo, MG7 Ab titers of the multi-epitope vaccine group were significantly higher than those of the mono-epitope group. Similarly, in mice treated with the multi-epitope vaccine, Ehlrich ascites carcinoma EAC xenografts were markedly smaller than tumors in the control animals [53].

Among other cancer-specific Abs whose antigen remains unknown, BCD-F9 is of potential interest because it is able to reduce the growth and metastasis of the human fibrosarcoma cell line HT-1080 in animal models. Screening a phage-displayed 10-mer library on this Ab allowed to isolate a peptide motif (12p) of sequence GRRPGGWWMR. Single-aa mutagenesis identified three residues as significant for the binding of 12p to BCD-F9 Ab. Rabbits immunized with this peptide produced specific antisera that inhibited the binding of BCD-F9 to HT-1080 cells in vitro and elicited antitumor activity against experimental metastasis in vivo [54].

In this same context, BAT is a monoclonal Ab that binds an uncharacterized 47–50 kDa membrane protein. The anti-tumor activity of BAT is mediated by its immunostimulatory properties, as demonstrated by adoptive transfer experiments in which splenocytes from BAT-treated mice induced regression of tumors in receiving mice [55]. By panning a phage-displayed peptide library on BAT, two mimotopes, Peptide A (PRRIKPRKIMLG) and B (QKILQQINLPRI), were isolated and characterized in competition assays on Daudi cells. In vivo, both peptides induced Abs capable of stimulating peripheral blood lymphocyte proliferation and IFN-γ secretion. Despite the relatively low titers, these Abs induced significant cytolytic activity of lymphocytes against both natural killer (NK)-sensitive and NK-resistant tumor cells. Immunization prevented the development of lung foci in mice inoculated intravenously with B16 melanoma cells, similar to what was observed following BAT treatment [28]. A successive study demonstrated that location, morphology, and inflammation of the lung metastases were similar in all groups (untreated, BAT-treated, and peptide-immunized mice). So, it was concluded that peptide treatment eliminates circulating tumor cells rather than exerting a tumoricidal effect in the lungs [56].

### 2.6. Mimotopes of Tumor-Associated Carbohydrate Antigens

Cancer cells are known to bear aberrant glycosylation profiles that affect both local invasion and metastatic spreading. A unique advantage of these tumor-associated carbohydrate antigens (TACAs) is that multiple proteins and lipids on cancer cells are tagged by the same carbohydrate structure; thus, targeting TACAs has the potential to raise an immune response against a broad spectrum of antigens. Glycans, however, are among the most challenging targets for vaccine design because of their poor immunogenicity. Peptide mimotopes of carbohydrates (also called glycoreplicas) are potentially suitable to overcome this limitation [57]. One of the first attempts to identify carbohydrate mimotopes goes back to 1997, when a phage-displayed 15-mer library was employed to screen Abs AD117m and H11, recognizing the neutral glycosphingolipids lactotetraosylceramide (Lc4Cer) and its isomer neolactotetraosylceramide (nLc4Cer), respectively. The two independent screenings provided sequences sharing high similarity and one identical motif (RNVPPTFNDVYWIAF), suggesting strong similarity between the antigen-binding sites of these Abs. Corresponding synthetic peptides were produced and confirmed as functional mimotopes [58].

Another group of glycosphingolipids, gangliosides include one or more negatively-charged sialic acid residues (N-acetylneuraminic acid, NANA). Panning the anti-GD1α ganglioside Ab KA17 with a 15-mer phage display library provided four specific phage clones. One of the selected motifs, WHWRHRIPLQLAAGR, interacted specifically with GD1+ RAW117-H10 murine highly metastatic lymphoma cells and inhibited their adhesion to hepatic sinusoidal endothelial cells. In vivo, intravenous injection of WHWRHRIPLQLAAGR led to an almost complete inhibition of RAW117-H10 metastasis to lung and spleen, and ~50% inhibition of liver metastasis [59]. GD2 is another ganglioside with TACA-like properties since it is highly expressed on tumor cells while having a restricted distribution in normal tissues except for neurons, skin melanocytes, and peripheral pain fibers. Overexpression of GD2 on highly malignant tumors such as neuroblastoma, a mostly fatal neoplasm in pediatric oncology, is yet another reason why GD2-mimicking peptides are often searched. Panning the anti-GD2 Ab 14G2a with a 15-mer phage-displayed library, followed by molecular modeling and mutagenesis studies, led to the identification and characterization of peptide 47-LDA (EDPSHSLGLDVALFM) as a mimotope of GD2. Immunization of BALB/c mice with a plasmid encoding 47-LDA resulted in the production of GD2-reactive Abs, which triggered complement-dependent cytotoxicity in vitro and protected against subcutaneous growth of human GD2-positive MV3 melanoma cells in vivo [60]. The 47-LDA vaccine also activated potent CD8+ T-cell responses when delivered simultaneously or 1 day after challenge with GD2+ syngeneic NXS2 neuroblastoma cells. Adoptive immunotherapy with CD8+ T lymphocytes isolated from 47-LDA-immunized and cured mice, in combination with plasmid-derived interleukin 15 (IL-15) and IL-21, led to regression of NXS2 tumors and prolonged tumor-free survival [61]. Further studies showed that the anti-GD2 Ab 14G2a cross-reacts with a 105 kDa glycoprotein expressed by murine and human neuroblastoma and melanoma cells, identified as activated leukocyte cell adhesion molecule (ALCAM/CD166). Stable silencing of CD166 expression in GD2-negative Neuro2a cells by CD166-specific short-hairpin RNA (shRNA) not only decreased the reactivity of these cells against 14G2a Ab, but also abolished recognition by 47-LDA vaccine-induced CD8+ T lymphocytes [62,63].

Selective motifs have also been identified by panning a 12-mer library of general formula xCx_8_Cx on the anti-GD2 Ab 14G2a. Five peptides (#8, NCDLLTGPMLCV; #65, SCQSTRMDPNCW; #85, VCNPLTGALLCS; #94, RCNPNMEPPRCF; and #D, GCDALSGHLLCS) have proven to specifically inhibit the binding of 14G2a to the human neuroblastoma cell line IMR-32 [64]. Among other Abs developed against GD2, the chimeric monoclonal ch14.18 is of interest since it has been demonstrated to recognize GD2+ neuroblastoma cells in the body when injected intravenously. From a constrained 10-mer phage-displayed library challenged on ch14.18, thirteen peptides were isolated and their specificity was validated in competitive binding assays with the natural antigen [65]. The GD2-mimicking properties of two such peptides (MD, CDGGWLSKGSWC and MA, CGRLKMVPDLEC), expressed and purified as recombinant proteins, were further investigated in docking experiments with ch14.18. In addition, the effect of GD2 mimotope was evaluated in peptide- and DNA-vaccination protocols, showing that mice treated with DNA vaccines develop GD2-specific innate cellular immune responses and are protected from spontaneous liver metastases in a neuroblastoma model [66]. The same group also worked on mimotope optimization by systematic alteration of the aa sequence with the SPOT technology, which is based on the synthesis of a defined peptide library onto a solid phase. MA peptide was subjected to systematic replacement of each aa within the mimotope sequence, leading to identification of C3, a peptide variant with a 5.6-fold lower K_d_ value. C3-KLH vaccine specifically suppressed primary tumor and spontaneous metastasis in a syngeneic model of neuroblastoma. Moreover, it induced an anti-tumor immune response characterized by systemic circulation of anti-GD2 IgGs [67]. In a further work, a 15-mer phage-displayed library was panned on another GD2-specific Ab, ME361. Based on the ability to inhibit the binding of ME361 to GD2, two peptides (LDVVLAWRDGLSGAS and GVVWRYTAPVHLGDG) were coupled to either KLH or multiple antigenic peptide (MAP) and tested in immunization experiments. Vaccination with these peptides (i) induced production of anti-GD2 Abs, (ii) triggered a delayed-type hypersensitivity response, and (iii) protected against tumor challenge with GD2+ melanoma cells [68]. Similarly, the anti-GD3 Ab 4F6 was panned with a 15-mer phage-displayed library and four sequences (GD3P1, LAPPRPRSELVFLSV; GD3P2, PHFDSLLYPCELLGC; GD3P3, GLAPPDYAERFFLLS; and GD3P4, RHAYRSMAEWGFLYS) were selected. GD3P4 was chosen as the most efficient and tested in vivo, where a MAP version of this peptide triggered the production of anti-GD3 Abs [69].

A widely recognized TACA, the tumor-associated carbohydrate Thomsen–Friedenreich antigen (TF-Ag) is overexpressed on the surface of several types of tumor cells, where it contributes to cell adhesion and migration during the metastatic spreading. Panning the anti-TF-Ag JAA-F11 Ab with a 12-mer library led to the identification of five peptide motifs (D2, HIHGWKSPLSSL; B1, HHSHKTNLATTP; B6, GHPHYITHKPNR; C1, YPSLPVYHSLRS; and D1, MHKPWSGHMQVP). D2 was further investigated for its binding specificity in several in vitro assays. In vivo, all the peptides were able to induce the production of specific Abs, in support of their development as potential anti-cancer vaccines [70]. In a more recent work, a phage display panning was performed on UN1, a Ab against CD43 glycoforms specifically expressed in lymphoblastoid T-cell lines and solid tumors, included breast, colon, gastric, and lung carcinomas. A total of 153 clones were positive to the UN1 Ab, 28 of which bound the Ab with high affinity and were sequenced to identify 11 unique motifs. Two peptides (W15, TCKLLDECVPLW and G23, SFAATPHTCKLLDECVPLWPAEG) were further characterized. In immunization experiments, a phage displaying the G23 sequence (named 2/165 phagotope) was capable of inducing a specific Ab response in BALB/c mice [71].

All these studies, although with somehow contrasting results, suggest that TACA mimotopes may be valuable in the development of therapeutic or even prophylactic cancer vaccines. However, antigenic mimicry (i.e., competition for binding the same Ab) not always means that the response is cross-reactive with a glycan moiety. Because carbohydrates and peptides are very different biochemically, further studies will be necessary to translate these results to the clinic [57].

### 2.7. Conclusions for this Section

Mimotope-based vaccines appear promising. However, despite the encouraging results summarized here, to our knowledge only one peptide derived from phage display screenings is as yet being investigated in clinical trials. This peptide, of sequence WRYTAPVHLGD (P10s), is a rational evolution of GVVWRYTAPVHLGDG (P10original) described by Wondimu and colleagues [68] as a glycoreplica motif binding the anti-GD2 Ab ME361. P10s, conjugated to the Pan T lymphocyte peptide PADRE, was initially designed for treatment of high-risk/recurrent breast cancer in a cohort of six patients. Preliminary data showed that the Ab response induced by the vaccine may improve chemotherapy efficacy and affect metastatic lesions in these patients [72,73]. Presently, other clinical trials are recruiting patients to evaluate P10s-PADRE in advanced stage lung cancer, stage IV breast cancer, and triple negative breast cancer (Highlands Oncology Group, Fayetteville, Arkansas, United States and University of Arkansas for Medical Sciences, Little Rock, Arkansas, United States. Data from www.clinicaltrials.org). If these trials confirm the clinical benefit of vaccination with P10s-PADRE, many others will probably follow and bring to full potential the huge amount of preclinical data that are still waiting to be developed. A list of peptide mimotopes described in this section, with reference to the Ab panned and the peptide motif isolated, is provided in Table 1.

## 3. Phages as Nanocarriers for Anticancer Vaccines

In cancer vaccination, either a TAA or a TAA-mimic molecule (mimotope) is presented to the patient’s immune system to prime and/or boost an immune response. In the case of mimotopes, this response may be directed against a synthetic/soluble peptide (as the ones described in the previous section) or against a peptide genetically or chemically anchored to the phage surface. A study focused on how the humoral response is raised in this setting demonstrated that (i) conjugation of a peptide with a phage particle gives a better response than conjugation with another carrier, e.g., ovalbumin (OVA), and (ii) modification of the phage coat protein to decrease its complexity and immunogenicity retargets the immune response to the peptide [74]. A schematic representation of Ab-based responses triggered by either free or phage-displayed TAAs (or their mimotopes) is presented in Figure 1.

The use of phages as vaccine carriers, in addition, may provide some benefits in the induction of a cellular response due to the nature of phages themselves. Being recognized as foreign antigens, phage particles *per se* induce an immunogenic response in humans by activating the innate immune system and eliciting adaptive immunity [75]. Phage-displayed peptides are processed and exposed on the MHC, leading to stimulation of both CD4+ (MHC II) and CD8+ (MHC I) T lymphocytes and induction of strong cytotoxic responses, a key feature for vaccines in general and for anticancer vaccines in particular [76,77,78] (Figure 2).

When administered intravenously, phages extravasate from the circulation and penetrate into tissues including tumor masses, thus representing a convenient route for vaccine administration. Last but not least, studies suggest that phages are non-toxic to animals [79], and in some case also to humans (see, e.g., [80]), and therefore unlikely to cause adverse effects. Several phage-based vaccines have been proposed for use in diverse biomedical applications (reviewed in [81,82,83]). They include different phage strains (e.g., T4 and λ) in addition to the most widely employed display system (the M13-derived filamentous bacteriophage *fd*), with peculiar features and specific capabilities to stimulate the host’s immune system [84].

### 3.1. Phage Particles Displaying Antigenic Portions of Melanoma Antigen Gene and Related Proteins

Melanoma antigen gene (MAGE) family consists of several chromosome X-linked genes first identified as they encode tumor antigens recognized by CTLs. MAGE proteins are present in several types of human cancer but not in normal tissues, thus representing potentially valuable TAAs. An antigenic MAGE-A1 nonapeptide of sequence EADPTGFSY (MAGE-A1_161–169_) was expressed as a fusion with the *fd* major coat protein pVIII and the deriving hybrid virion was tested for its capacity to induce immune responses. In vitro, both wild-type and MAGE-A1_161–169_-phages elicited stronger delayed time hypersensitivity responses and enhanced NK cell activity. However, only splenocytes derived from mice immunized with MAGE-A1_161–169_-phages (but not from those immunized with wild-type phages) exhibited lytic activity against B16-F10 murine skin melanoma cells. *In vivo*, two trials were performed to determine whether MAGE-A1_161–169_-phages can (i) protect against tumor grafting (prevention) and/or (ii) inhibit the growth of established tumors (immunotherapy). For these prevention and immunotherapy trials, C57/BL6J mice were injected with the phage vaccine a few weeks prior to, or 5 days after, subcutaneous implant of B16-F10 cells, respectively. In both cases, vaccination with MAGE-A1_161–169_-phages resulted in controlled tumor growth and prolonged overall survival compared with wild-type phage [85]. Similarly, the work from Sartorious and colleagues describes an antigen delivery system in which p23 (KDSWTVNDIQKLVGK) (a promiscuous HLA-DR-restricted T-helper peptide) and either MAGE-A10_254–262_ (GLYDGMEHL) or MAGE-A3_271–279_ (FLWGPRALV) (both HLA-A2-restricted TAAs) are fused to the f*d* phage pVIII protein, creating the hybrid virions fd23/Mg10 and fd23/Mg3, respectively. In vitro, peripheral blood mononuclear cells (PBMCs)–used as a model of APCs—induced specific T-lymphocyte cytotoxic responses when pulsed with fd23/Mg10. Treatment with wild-type phages in the presence of soluble p23 and MAGE-A10_254-262_ had no effect, suggesting that a complete virion is necessary to induce cytotoxicity. Similarly, a cytotoxic activity was observed in splenocytes from mice immunized with either fd23/Mg10 or fd23/Mg3. In vivo, HHD transgenic mice [86] were vaccinated with fd23/Mg3 and then challenged with syngeneic EL-4-HHD lymphoma cells overexpressing MAGE-A3. At day 21, a tumor was present in 100% of control mice, but only in 36% of fd23/Mg3-immunized mice. At day 80, 60% of fd23/Mg3-immunized mice were still alive, while all control mice died between day 34 and 60 [87]. In another work, circulating Abs from a breast cancer patient were immobilized on protein G-coupled magnetic beads and used to immunocapture potential TAAs, retrieving MAGE-1 among others. Oral immunization of BALB/c mice with T7 phages exposing a selection of identified TAAs (MAGE-1-T7, SSX2-T7, or p53-T7) induced effective Th1 and Th2 responses against the antigens. Another selected TAA (Hsp27) was expressed as a fusion with the capsid protein of phage T7 and the deriving hybrid vector was used to immunize BALB/c mice, followed by subcutaneous implant of syngeneic 4T1 breast adenocarcinoma cells. Average tumor weights and metastasis number in mice immunized with Hsp27-T7 were significantly lower than in the controls, supporting a preclinical efficacy of this T7 phage-based vaccine [88]. A murine gene sharing several characteristics with human MAGEs, P1A was identified as encoding a protein specifically recognized by a clone of CD8+ T lymphocytes raised against the murine mastocytoma P815 model. P1A includes a nonapeptide (LPYLGWLVF, P1A_35–43_) that binds to the MHC I determinant H-2d to generate antigen P815AB, a target for immune responses against tumor P815. Hybrid *fd*-derived virions displaying P1A_35–43_ and a 6x-histidine tag fused to the pVIII protein (pC89hisP1A) were prepared. Splenocytes from mice immunized with pC89hisP1A exhibited specific anti-P1A cytotoxic responses. T lymphocytes from the same mice produced high levels of IFN-γ, mostly (75%) released by CD4+ cells, revealing the generation of Th1-dominated immune responses. In an in vivo prevention trial, pC89hisP1A was administered subcutaneously to DBA/2 mice prior to implant of P815-F3 cells. While no control animal survived after day 60, 70% of pC89hisP1A-immunized mice were still alive, demonstrating prophylactic efficacy. Similarly, in an immunotherapy trial DBA/2 mice challenged with pC89hisP1A four days after implant of P815-F3 cells showed prolonged survival and delayed tumor growth (with 2 out of 30 mice being cured), in support of a therapeutic efficacy of the vaccine [89]. Despite the promising results, to our knowledge there has not been any follow up to such preclinical studies.

### 3.2. Phage Particles Displaying Antigenic Portions of HER2

As described in the previous section, HER2 has been the subject of several studies involving phage display in the context of immunotherapy. Some specific sequences of HER2 are sufficiently immunogenic to stimulate CTLs to recognize and kill HER2+ cancer cells, and have therefore been investigated in both preclinical and clinical settings. Recent papers from Behravan group describe the engineering and testing of phage-based vaccines that include one of two HER2-derived peptides, E75 (HER2_369–377_, KIFGSLAFL) [90] and AE37 (HER2_776–790_, GVGSPYVSRLLGICL) [91], previously investigated as candidate vaccines in clinical trials for breast, ovarian, and prostate cancer [92,93,94,95,96,97,98,99]. E75 peptide was exposed on λ phage particles as a fusion with the gpD coat protein and the deriving virions λF7(gpD::E75) were used to immunize BALB/c mice. High levels of IL-4 (associated to humoral immunity) and IFN-γ (associated to cellular immunity), as well as increased levels of CD8+ cells, were detected in sera from immunized mice. Splenocytes from the same mice showed specific cytotoxic activity against HER2+ TUBO mouse mammary tumor cells in vitro. Both preventive and curative effects were investigated in vivo in BALB/c mice implanted with TUBO cells. While λF7(gpD::E75) failed in the prevention trial, the curative trial was somehow successful in delaying tumor growth [90]. In the other work, the same experiments were repeated with λ phages exposing AE37 [λF7(gpD::AE37)], obtaining similar in vitro results. In vivo, λF7(gpD::AE37) was more efficient than λF7 (gpD::E75): not only curative but also prophylactic effects were observed [91].

### 3.3. Phage Particles Displaying Antigenic Portions of Mucin 1

The transmembrane protein mucin 1 (MUC1) is a functional target for anticancer approaches and a likely valuable TAA for immunotherapy. In normal cells, MUC1 is heavily glycosylated, so its antigenic epitopes are shielded from the immune system; in contrast, MUC1 is under-glycosylated and overexpressed (~100-fold) on the surface of cancer cells. The extracellular domain of MUC1 consists of tandem repeats of a 20-aa motif (PDTRPAPGSTAPPAHGVTSA), where the serine and threonine residues represent anchor points for O-glycosylation. In cancer cells, the threonine close to the C-terminal of the motif is linked to N-acetyl-galactosamine creating a TAA referred to as Tn antigen. Phage Qβ has been exploited to generate non-infective virions known as virus-like particles (VLPs), which can be chemically conjugated with peptides or even with large and complex proteins [100]. VLPs carrying antigenic epitopes have proven to be immunogenic, thus representing good candidates for vaccine development. In a recent work by Yin and colleagues [101], MUC1 peptides with different glycosylation patterns were linked to Qβ phage, the deriving Qβ-MUC1 conjugates were administered to C57/BL6 mice, and the induction of specific Abs was investigated. Huge titers of Abs were generated, with broad selectivity for different glycosylation patterns of MUC1 as demonstrated by ELISA and microarray analysis. These Abs were able to bind MUC1-expressing tumor cells and kill them by complement-mediated cytotoxicity. Splenocytes and lymph node cells from mice immunized with Qβ-MUC1 induced CTL responses both in vitro and in vivo [101].

### 3.4. Phage Particles Displaying Antigenic Portions of VEGFR2

On the one hand, self-antigens in tumor masses tend to be quantitatively (overexpression) more than qualitatively (mutation) different from those in normal tissues, and this feature leads to immune tolerance against the TAA. On the other hand, because of potential induction of self-immunity, targeting those antigens has been mostly neglected until recently [102]. Despite these drawbacks, immunotherapy directed to self-antigens is now emerging as a promising approach in cancer therapy. VEGFR2 is the major receptor for VEGF and mediates numerous signal pathways in angiogenesis, such as microvascular permeability, endothelial cell proliferation, invasion, migration, and survival. VEGFR2 is often overexpressed in tumor endothelial cells, therefore representing a potentially suitable TAA. A VEGFR2-based vaccine was created by displaying the extracellular portion of mouse VEGFR2 protein (mVEGFR2) on the surface of phage T4. ELISA and Dot ELISA assays confirmed the presence of anti-mVEGFR2 Abs in the sera of immunized mice, which were able to inhibit capillary-like tube formation in an in vitro angiogenic assay with human umbilical vascular endothelial cells (HUVECs). In vivo, both prophylactic immunization with T4-mVEGFR2 and adoptive transfer of purified Abs from T4-mVEGFR2-immunized mice to Lewis lung carcinoma-bearing mice resulted in inhibition of tumor growth and prolonged survival. Further results suggest that the T4-mVEGFR2 vaccine exerts its anti-tumor activity mainly through CD4+ T lymphocytes. In the discussion, the authors speculate that using T4 as a vaccine carrier may circumvent the immune tolerance to VEGFR2 thanks to the high immunogenicity of the viral coat proteins [103].

### 3.5. Phage Particles Exposing Phage Display-Derived TAA Mimotopes (Miscellaneous)

As described previously, phage-displayed libraries are screened to isolate mimotopes—peptides that mimic the secondary structure and antigenic properties of a protein, carbohydrate, or lipid. Because such mimotopes are selected as the best fit for specific anti-TAA Abs, they are expected to be at least as efficient as, and likely more than, natural TAA epitopes. For example, the EM.L2 peptide (QHYNIVNTQSRV), previously identified as an EGFR mimotope specific for the IRC-62 Ab [40], was expressed as a fusion with the L2 extracellular domain of human EGFR, a 6x His tag, and the pVIII protein of *fd* phage. The deriving peptide-displaying phages were tested in a mouse model of Lewis lung carcinoma, confirming the induction of specific anti-EM.L2 Abs. Following vaccination with EM.L2-displaying phages, both humoral and cellular responses were activated, as demonstrated by increased levels of IL-4 and IFN-λ. In vivo, the EM.L2-displaying phage was tested as both a prophylactic and therapeutic vaccine in a model of Lewis lung carcinoma. In both settings, tumor growth was inhibited, although not fully blocked, in the prophylactic trial [104].

In another work, a phage display panning was designed to identify new immunogenic and tumor-targeting peptides. Panning was performed firstly in vivo in C57BL/6 mice bearing B16-F10 syngeneic skin melanoma tumors (a highly aggressive and poorly immunogenic model), followed by additional rounds of selection on suspended B16-F10 cells in vitro. A phage designated WDC-2 (displaying the TRTKLPRLHLQS peptide motif) proved to be a specific ligand of B16-F10 cells and was therefore chosen for further investigation. When tumors reached a palpable volume, mice were injected peritumorally with either the WDC-2 phage or a control (PBS or a non-specific Fab-exposing phage). No mice survived more than 18 days in the control groups; in contrast, 20% of the mice immunized with WDC-2 phage survived up to 30 days. In other experiments performed in B16/A2Kb tumor-bearing mice, WDC-2 phage was compared with an HLA-A2-specific Fab-phage, observing complete regression of established tumors and long-term survival in 42 or 53% of mice, respectively. Immunohistochemistry of tumor sections revealed massive neutrophil infiltration within 24 h from injection. In vitro, phages induced IL-12 and IFN-γ production in mouse splenocyte cultures. In addition to these immune responses, the authors suggest that an unspecific antiviral inflammation may be triggered by phage treatment, thus contributing to tumor regression [79].

In a study by Samoylov and colleagues, phage display was used to identify mimotopes of the gonadotropin releasing hormone (GnRH), with the intent to stimulate an endogenous production of therapeutic Abs. Polyclonal anti-GnRH Abs (either commercial or purified from cat and dog sera) were subjected to panning with a 8-mer library expressed as a fusion with the pVIII protein of *fd* phage. Clones displaying potential GnRH mimotopes (peptide motifs: EGLRPSGQ, DGLRPQAP, EHPSYGLA, EPTSHWSA, and DAPAHWSQ) were chosen based on their sequence similarities to GnRH and tested for their capability to induce the production of specific Abs in mice. Among these, EHPSYGLA was identified as the motif triggering the most long-lasting humoral responses, although in this preliminary trial no functional outcome (i.e., suppression of testosterone levels in the blood) was observed. Additional in vivo studies were performed with EHPSYGLA-phage either at higher doses or in combination with a a poly(lactide-co-glycolide) (PLGA)-based adjuvant, demonstrating a multifold increase in Ab production paralleled by a substantial decrease in the production of testosterone. Immunization of mice with EHPSYGLA-phage, even at the highest doses, resulted in no adverse effect. While the main purpose of this research group remains animal contraception, an approach similar to the one described in this paper—as suggested by the authors themselves—might be useful in the treatment of human hormone-sensitive cancers of the reproductive trait [105].

### 3.6. Clinical Trials with Phage-Based Anticancer Vaccines

As of today, entering “phage” in the American database of clinical trials (www.clinicaltrials.gov) returns 42 records, most of which are focused on infectious diseases. Current research mainly explores the lytic property of phages and their proteins to develop treatments for antibiotic-resistant bacterial infections [106]. Only one study (NCT02757755, concluded in 2016) included phage as the main component, but the scope was a mere biosafety evaluation. A single report of a European phase I/II clinical trial on patients treated with a phage-based vaccine (MKI-TR02/2002) was published in 2014 [107]. The trial follows a preclinical study in which a selected B-cell receptor—Ig variable region peculiarly exposed by each malignant B-cell clone, also called idiotype—was chemically linked to the surface of phage particles. The deriving Id-phages were tested in immunization experiments in BALB/c mice either tumor-free (tolerance and dosage experiments) or bearing syngeneic BCL1 lymphomas (efficacy studies), demonstrating both safety and tumor protection [108]. Following preclinical toxicity studies in mice and rabbits, the Id-phage was also evaluated as a potential vaccine in patients. Fifteen adults with terminal-stage multiple myeloma were subjected to a total of six intradermal immunizations (day 1, 7, 14, week 4, 8, 12), with the Id-phage vaccine in the presence of KLH as an immunogenic carrier and GM-CSF as an immunostimulatory adjuvant. The initial Id-phage dose of 0.25 mg for patients 1–5 was subsequently escalated to 1.25 mg for patients 6–10 and 2.5 mg for patients 11–15, as no serious adverse event occurred. Phage vaccination was well-tolerated with only minor and transient side effects, such as skin irritation at the injection site and flu-like symptoms. The efficacy of vaccine treatment was evaluated as decrease in serum paraprotein (M level) and urine-excreted myeloma-specific light chains, followed by characterization of humoral (titer of anti-idiotype Abs) and cellular responses (CTL and delayed type hypersensitivity). Patients exhibiting a clinical response to Id-phage vaccine also produced anti-idiotype Abs. Due to high tumor load and impaired immune function, these patients were not expected to respond to active immunotherapy; nevertheless, the Id-phage vaccine induced a clinical response in most of them. Further studies with a larger cohort of patients will be necessary to confirm these data and confer statistical significance to these potentially interesting results [107].

### 3.7. Conclusions for this Section

Phages exert immunomodulatory effects and may be efficiently exploited in the design of clinical applications. However, the continuous/repeated administration of phages is expected to induce memory cells with subsequent generation of anti-phage Abs. So, it is generally accepted that the utilization of phages either for the production of new therapeutics, as gene delivery vehicles, or as tumor targeting agents (at large, for any clinical purpose) must be carefully evaluated as per their immunogenicity and retention time [75]. Among potential drawbacks, circulating phages are inactivated by macrophages of the reticuloendothelial system in synergy with anti-phage Abs. In addition, while phage preparations administered *per os* may downregulate the antigen processing abilities of intestinal dendritic cells, phages administered intravenously are rapidly cleared by liver cells [109,110]. There is a general lack of knowledge as to the fate of phages in the human body in terms of cellular interactions [111]. For example, the presence of circulating phages (“phagemia”) may have as yet unknown functional consequences. To our knowledge, there are only two reports confirming a natural phage occurrence in human sera (i.e., unrelated to their exogenous administration) [112,113]. Another example is phage uptake after oral administration, which requires more study since the few results available are discordant. The presence of phages within the gastrointestinal tract, often in significant numbers, raises the possibility of their interactions with enterocytes and gut-associated lymphoid tissue, an immunological network that includes the majority of T lymphocytes and a significant proportion of B lymphocytes [109]. Most of all, the influence of phages on the immune system remains poorly understood. A list of the phage-based vaccines described in this section, with reference to the antigenic epitope displayed, the phage carrier, the peptide motif, and the animal model investigated, is provided in Table 2.

## 4. Phage Display-Derived Peptides as Nanomodulators of the Immune Response

As small molecules easy to characterize and produce on a large scale, peptides are promising tools for targeted approaches in cancer immunotherapy. Initial studies investigated the use of peptides to stimulate the immune system indirectly. Successive reports focus on two main applications: interfering with immune checkpoints and modulating the activity of immune cells. These approaches are schematized in Figure 3.

### 4.1. Targeting the Immune System Indirectly

Early applications of phage display-derived peptides to stimulate the immune system against cancer cells started with the search for ligands of the IgM λ receptor on lymphoma cells. This receptor is unique to each patient, therefore the use of therapeutic anti-idiotypes, despite initially promising, has proven unpractical because the selection and optimization of monoclonal Abs remain time- and money-consuming procedures. As an alternative, by panning three phage-displayed libraries (based on random 8- or 12-mer motifs) on the IgM λ receptor expressed by the human Burkitt lymphoma cell line SUP-B8, Renschler and colleagues isolated several clones sharing one of the four consensus sequences KP--xRV, -W--WxR, Y--EDLRRR, and --PVDHGL. The first three were optimized by mutagenesis and the specificity of derived peptides (named A, B, and C) was confirmed by ELISA. Corresponding polymers of biotin-peptides bound to streptavidin induced specific cytotoxicity of SUP-B8 cells, paralleled by intracellular protein phosphorylation signaling [114]. In a successive study, the same group explored the use of these peptides for immunotherapy of B-cell lymphoma. Activation of SUP-B8 cells was quantified as acidification of the extracellular medium. In these assays, peptide B (DWAIWSKRGGK) and C (YSFEDLYRRGGK) showed comparable efficacy, being peptide C active at slightly lower doses. This was the first evidence that B-lymphoma cells can be activated by peptide ligands in addition to anti-idiotype Abs [115].

In another set of experiments, melanin was chosen as a target. This protein is released by melanocytes with altered membrane permeability or undergoing cell death (such as in melanomas) and so it represents a potential TAA that can be accessed through the blood circulation. By panning a 7-mer library on recombinant melanin, twenty-four individual clones were identified, eight of which proved to be specific binders and three (HTTHHRN, TTHQFPF, and NPNWGPR) were further investigated. Each peptide was individually linked to ^188^Re via the crosslinker hydrazinonicotinamide (HYNIC) and validated for selective binding to the highly melanized human melanoma cell line MNT1. In vivo, administration of ^188^Re-HYNIC-NPNWGPR to A2058 human metastatic melanoma-bearing nude mice resulted in retardation of tumor growth. The unlabeled version of NPNWGPR was almost as efficient as the ^188^Re-HYNIC–peptide, suggesting that the observed therapeutic effect was independent from the attached radionuclide. The work led to the speculation that melanin-specific peptides are absorbed by melanin on the cell surface, where they become immunogenic and trigger a localized inflammation responsible for the antitumor effect [116].

### 4.2. Targeting the PD-1/PD-L1 Axis with Peptide Inhibitors

One of the most characterized immune checkpoint pathways, the PD-1/PD-1L axis is hyper-activated by cancer cells to evade the immune surveillance [5]. Several therapeutic Abs that inhibit either PD-1 or PD-1L have been tested in clinical trials and some of them (namely, pembrolizumab, nivolumab, atezolizumab, avelumab, durvalumab, and cemiplimab) are FDA-approved for different types of cancer. However, less expensive treatments alternative to monoclonal Abs (e.g., peptides) are continuously sought. In a first study, mirror-image phage display was employed. In this protocol, a D-version (mirror image) of a natural target protein (composed of L aa) is produced by chemical synthesis and used to screen L-peptide libraries displayed on phages. D-versions of the selected L-peptides are then synthesized, which, by symmetry, are expected to bind to the target protein. Such D-peptides usually exhibit better stability in vivo than their L-counterparts so they are preferred for some applications. A D-version of PD-L1 IgV domain was used as a bait for a 12-mer library produced on *fd* phages. The two most represented selected motifs, PPA-1 (NYSKPTDRQYHF) and PPA-2 (KHAHHTHNLRLP) were synthesized as D-peptides and their specificity was confirmed by surface plasmon resonance. In competition experiments, both peptides blocked PD-1/PD-L1 interaction at the cellular level, as evaluated by flow cytometry. In vivo, both peritumoral and intraperitoneal injection of ^D^PPA-1 inhibited the growth of syngeneic CT26 colon carcinomas in BALB/c mice. Biodistribution analysis showed efficient delivery of the peptide, with specific accumulation in the tumor mass and background uptake by the liver, kidney, stomach, and lungs [117]. A bacterial surface display library was used by Li and colleagues to identify PD-L1-binding peptides, obtaining several clones with the consensus sequence CWCWR. A focused library with the format x_5_CWCWRx_5_ was then prepared and screened. The best candidate, TPP-1 (SGQYASYHCWCWRDPGRSGGSK), (i) bound to human PD-L1 specifically and with high affinity, (ii) reversed PD-L1-mediated inhibition of T lymphocyte activation in terms of proliferation and IFN-γ release, and (iii) delayed tumor growth in a xenograft model, with effects comparable to the FDA-approved Ab durvalumab [118]. In another work, a truncated version of PD-1 bearing the mutation C93S was displayed as a fusion with the pIII of *fd* phage and used as a template to construct a library by site-specific mutagenesis of 24 aa. The deriving phage-displayed library was panned on human PD-L1-biotin bound to streptavidin beads, leading to the isolation of PD-1 variant L5B7 (VNYMSNQTKAPPGLSAILPYIQIE) with 3000-fold affinity increase as evaluated by in vitro binding assays on recombinant PD-L1. PBMCs stimulated with anti-CD3 and anti-CD28 agonist Abs were used as a T-lymphocyte activation model to evaluate the efficacy of L5B7. In these assays, L5B7 increased both PBMC proliferation and IFN-γ release by an extent comparable to soluble human PD-1 and anti-PD-1 Ab [119]. A couple of analogous reports were presented at the 2019 meeting of the American Association for Cancer Research (AACR). The first one describes peptide CLP002 as a specific ligand of PD-L1 with promising activity in vitro and in vivo in CT26 tumor-bearing mice [120]. The second one (by Avacta Life Sciences, Ltd.) describes Affimer^®^ biotherapeutic antagonists—monomeric scaffold proteins based on the human protease inhibitor Stefin A—to human and mouse PD-L1, again with encouraging preclinical activity [121].

Finally, in a recent work, Son and colleagues designed a phage-displayed library based on a small-sized non-Ab scaffold, termed “repebody”, to identify ligands of human PD-1. A clone with high affinity (r_A1) was selected and further maturated up to a K_d_ = 17 nM by modular evolution. The deriving molecule (r_G9) was validated both in vitro (specific binding in competition assays and flow cytometry, restoration of T lymphocyte activity) and in vivo (pharmacokinetics and antitumor effect in mice xenografted with NCI-H292 human lung cancer cells) with positive results [122].

### 4.3. Modulating the Activity of Immune Cells

#### 4.3.1. Inducing Immune Responses by Targeting APCs and T Lymphocytes

A few reports have been published that describe phage display-selected peptide ligands of molecules expressed by specific immune cell lineages. One such molecule is CD11c/CD18 (integrin α_X_β_2_), which is present exclusively on the surface of APCs, particularly monocytes and dendritic cells. In a first study, a cyclic 10-mer (CL10) and a linear 9-mer (LL9) libraries were panned on CD11c/CD18 purified from the spleen of a patient with hairy cell leukemia. The circular peptide motif CGRWSGWPADLC was isolated as a ligand of CD11c/CD18 on monocytes, as established by in vitro binding assays, surface plasmon resonance, and flow cytometry. This motif shares similarity with both the linear GNWTWP epitope in domain D5 and the phase-shifted PEDNGRSFS epitope in domain D4 of Intercellular Adhesion Molecule 1 (ICAM-1), the molecular partner of CD11c/CD18. Binding assays with deletion mutants of ICAM-1 mapped the binding region to domain D4 [123]. In a successive study, this same peptide (p30) plus two additional peptides derived from the CD11c/CD18-interacting portion of ICAM-4 (p17 and p18) were individually fused to a 12x histidine tag and crosslinked to stealth liposomes. Specific binding to human monocyte-derived dendritic cells, the murine dendritic cell line DC2.4, and CD11c+ murine splenocytes was validated by flow cytometry. In vivo, tracer-embedding liposomes, engrafted with the CD11c/CD18-binding peptides and administered intravenously, accumulated specifically in CD11c+ splenocytes. The potential of CD11c-targeting liposomes to induce an immune response was first evaluated in healthy C57BL/6 mice, observing (i) an increase in splenocyte proliferation (30- to 59-fold, depending on the displayed peptide), (ii) specific expansion of IFN-γ-producing CD8+ T lymphocytes, and (iii) production of high-titer IgG_1_. Tumor cell-derived plasma membrane vesicles (PMV) grafted with the CD11c-targeting peptides (p18-PMV and p30-PMV) were evaluated for their efficacy as prophylactic and/or therapeutic vaccine in mice bearing subcutaneous syngeneic melanoma B16-OVA cells. In both settings, treatment with pCD11c-PMV led to a marked reduction in the number of lung metastases and regression of the primary tumor, which was complete in some cases [124].

Another marker highly expressed on the surface of dendritic cells is the C-type lectin Clec9a, a protein involved in antigen uptake and induction of humoral immunity. A 12-mer phage displayed library was used to screen murine Clec9a C-type lectin-like domain fragment, retrieving four motifs, one of which (peptide WH of sequence WPRFHSSVFHTH) was further validated by in silico and in vitro studies. An OVA_257–264_-conjugated version of WH increased IFN-γ secretion by Clec9a+ dendritic cells and promoted antigen-specific CTL induction, measured as increased mRNA levels of both IFN-γ and cytotoxicity markers (perforin and granzyme B). In vivo, OVA_257–264_-WH was evaluated as a potential vaccine in the B16-OVA lung metastasis model, observing a marked decrease in the numbers of lung foci, paralleled by induction of IFN-γ and cytotoxicity markers in splenocytes, at a higher extent compared with OVA alone [125]. Both CD11c/CD18 and Clec9a have a functional role in the first phases of the immune response, namely antigen presenting and T-lymphocyte priming, therefore the targeting peptides may have a direct effect in addition to function as a vaccine.

Phage display-derived peptides have also found applications in later stages of the immune response involving the action of T lymphocytes. In a pioneering work, Kraft and colleagues panned a 12-mer library on recombinant α_v_β_6_ integrin expressed as a transmembrane truncated soluble receptor, retrieving the RGD motif in 51% of the clones and the DLxxLx motif in 27%, the latter proving to be highly specific for α_v_β_6_ [126]. From the clones selected in this study, the peptide motif RTDLDSLRTYTL (Bpep) was exploited in a successive work to serve as a Chimeric Antigen Receptor (CAR) targeting domain. Primary human T lymphocytes were genetically modified to express the Bpep-CAR, consisting of the α_v_β_6_-binding peptide followed by a triple glycine linker, an IgG_4_ hinge region, the CD4 transmembrane domain, and the cytoplasmic signaling domain of CD3-ξ. In vitro, the Bpep-CAR-exposing lymphocytes recognized and killed the human ovarian cancer cell line OVCAR-3 and two primary ovarian tumor lines (RSN001 and RSN002), all positive for α_v_β_6_ integrin. Upon engagement with α_v_β_6_ integrin, Bpep-CAR-expressing T lymphocytes were activated to release high levels of IFN-γ [127]. Since many tumor-targeting peptides have been identified by phage display, it is expected to see more applications in the field of CAR T cells in the near future.

Another subpopulation of effector cells, T lymphocytes bearing the γδTCR, show promise in immunotherapy because of their potent cytotoxicity toward various types of tumor cells. A panel of complementary determining regions (CDR3δ) derived from ovarian epithelial carcinoma infiltrating γδT lymphocytes was screened with a 12-mer phage library obtaining seven putative epitopes (EP1, WPHNWWPHFKVK; EP2, PLLPMHPMKVSH; EP3, KPPTQKRRRQTM; EP4, RPRTRLHTHRNR; EP5, YPWHWWHSVSPW; EP6, FHWSWYTPSRPS; and EP7, WHHPWWYPRPGV). The following tandem epitopes were constructed as GTS fusion proteins: EPt4 (EP1-EP5-EP6-EP7), EPt7 (EP1-EP2-EP3-EP4-EP5-EP6-EP7) and EPt8 (EP1-EP5-EP6-EP7-EP1-EP5-EP6-EP7). In vitro, the tandem peptides (i) activated γδT lymphocytes by inducing proliferation and release of IL-2, IFN-γ, TNF-α, FasL, and granzyme B and (ii) triggered tumor cell cytotoxicity, being GST-EPt7 the most active. In BALB/c nude mice carrying human ovarian cancer SKOV3 cells, co-administration of GST-EPt7-preactivated γδT lymphocytes and IL-2 resulted in smaller tumors and extended survival [128].

#### 4.3.2. Reducing Immunosuppression by Targeting Tumor-Associated Macrophages and Treg Cells

Other immune cells such as tumor-associated macrophages (TAMs) impact on cancer growth by influencing the adaptive immune responses. Depending on different exogenous stimuli, TAMs are found in two polarization states with opposite functions: IFN-γ and lipopolysaccharide (LPS) trigger a proinflammatory phenotype (M1-like), while IL-4 and IL-13 induce an anti-inflammatory phenotype (M2-like) characterized by poor antigen presentation, stimulation of angiogenesis, and tissue remodeling. The number of infiltrating M2 TAMs correlates with tumor progression, drug resistance, and poor survival. In this case, it is desirable to inhibit their activity and/or selectively deplete them from the tumor microenvironment. Cieslewicz and colleagues designed a subtractive phage panning strategy using in vitro-activated subpopulations of M1 and M2 macrophages. An M2-specific peptide, M2pep (YEQDPWGVKWWY), was isolated and validated against mixed populations of primary cells. M2pep was fused to the pro-apoptotic peptidomimetic _D_(KLAKLAK)_2_ (KLA) [129], and the efficacy of the deriving conjugate was evaluated in vitro and in vivo. Following intravenous administration of M2pep-KLA, BALB/c mice bearing the CT26 model showed prolonged survival rates and slower-growing tumors with reduced infiltration of M2-like TAMs [130].

Tregs are immunosuppressive cells whose function is often exploited by tumor cells to evade the immune responses. Few approaches have been designed to inactivate Tregs from the tumor microenvironment, but with diverse outcomes due to an overall lack of specificity. Moreover, Tregs do not expose cell-surface markers—the most specific marker is Foxp3, an intracellular transcription factor—so they are quite difficult to target. By screening a phage-displayed library, Casares and colleagues identified a 15-mer synthetic peptide (P60, of sequence RDFQSFRKMWPFFAM) [131], which was able to enter the cell, bind to Foxp3 and inhibit its nuclear translocation. As the peptide can penetrate any cell in the body, the amount required to exhibit an anti-tumoral effect would be high, limiting the clinical feasibility of this approach. Therefore, P60 was fused to an aptamer to obtain the conjugate CD28Apt-P60 with double targeting properties (CD28-expressing cells and Fox3). In vitro, CD28Apt-P60 inhibited Treg activity more efficiently than the unconjugated peptide. In vivo, systemic administration of CD28Apt-P60 resulted in improved efficacy of an OVA-based vaccine and reduction of tumor burden in the CT26 model in a prophylactic setting [132].

### 4.4. Conclusions for this Section

Phage display-selected peptides can be exploited as modulators of immune cell activity in terms of stimulation of effector cells (APCs, lymphocytes) and/or inhibition of suppressor cells (TAMs, Tregs). Because these approaches have demonstrated good efficacy and specificity coupled with low or absent toxicity, they may offer a good alternative to Ab-based immunotherapy. The growing evidence available in the literature supports the feasibility of these applications, which are likely to be taken to clinical studies soon. A list of the phage display-derived peptide modulators of the immune system described in this section, with reference to the target molecule/cell type and specific effect, is provided in Table 3.

## 5. Final Remarks

Phage display is a powerful nanotechnology with broad applications. In the rising field of tumor immunology, phage display-selected peptides can be included in a therapeutic regimen as mimotopes (prophylactic and/or therapeutic vaccines) or used as small-molecule effectors (activators or inhibitors of immune cells, effectors of immune checkpoint molecules). In addition, phages themselves are suitable carriers for peptide and protein vaccines due to their immunogenicity coupled with low toxicity, although more studies are needed to fully clarify their interaction with the mammalian host. As yet, there are no commercial phage-derived medicines for cancer immunotherapy [133,134]. This limitation arises primarily from the difficulties associated with peptide stability and delivery, phage immunogenicity, and from the challenges posed by the diversity of human immunogenetics [84]. Nevertheless, such nanotechnological systems, supported by state-of-the art instrumentation and bioinformatics, have the potential to represent innovative and efficient tools for personalized application in the near future.

## Figures and Tables

**Figure 1 molecules-25-00843-f001:**
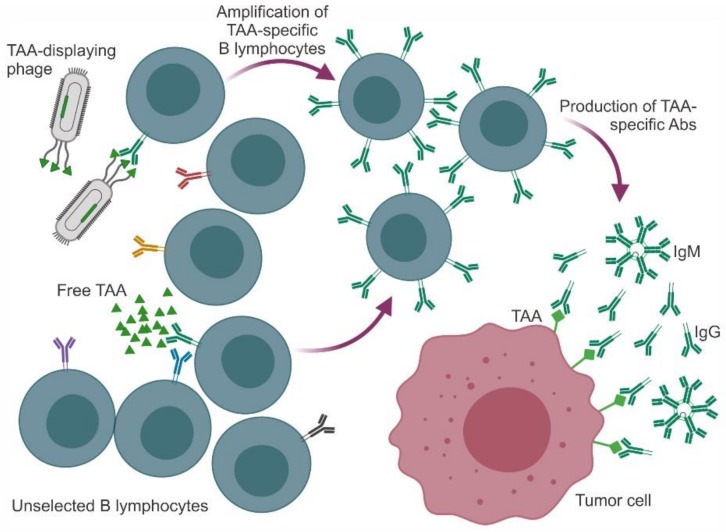
Humoral response: B lymphocytes. Both TAA-mimic peptides and TAA-displaying phage particles are recognized by B lymphocytes, which respond by producing high-affinity, anti-TAA Abs. These Abs are released and enter the bloodstream to reach tumor cells and tag them for neutralization and/or destruction. Abbreviations: TAA, tumor-associated antigens.

**Figure 2 molecules-25-00843-f002:**
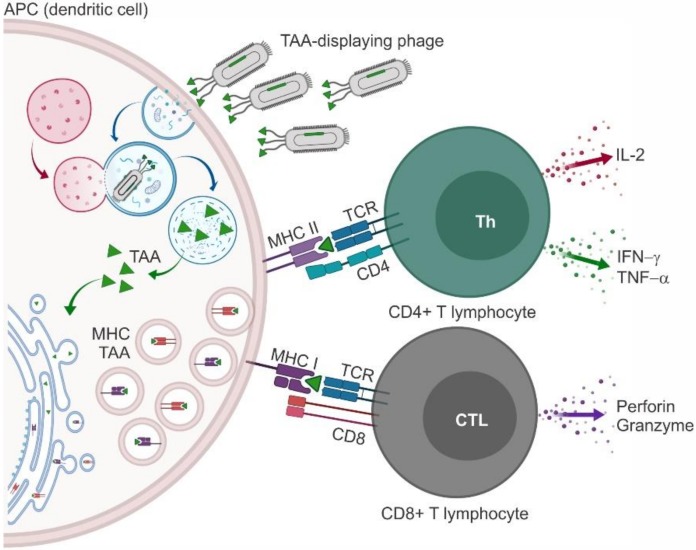
Cellular response: APCs and T-lymphocytes. TAA-displaying phages are processed inside an APC via the endosome-lysosome pathway. The TAA is exposed on the APC surface complexed with MHC molecules. Naïve T lymphocytes carrying a TAA-specific TCR are activated by interaction with the TAA-MHC complex: binding to MHC class II leads to activation of CD4+ T lymphocytes and differentiation in Th cells, while binding to MHC class I leads to activation of CD8+ T lymphocytes and differentiation in CTLs. Both types of effector T cells release chemokines and cytokines to mount an immune response against the TAA. Abbreviations: APC, antigen-presenting cell; TAA, tumor-associated antigen; MHC, major histocompatibility complex; TCR, T-cell receptor; Th, T helper; CTL, cytotoxic T lymphocyte; IL-2, interleukin 2; IFN-γ, interferon γ; TNF-α, tumor necrosis factor α.

**Figure 3 molecules-25-00843-f003:**
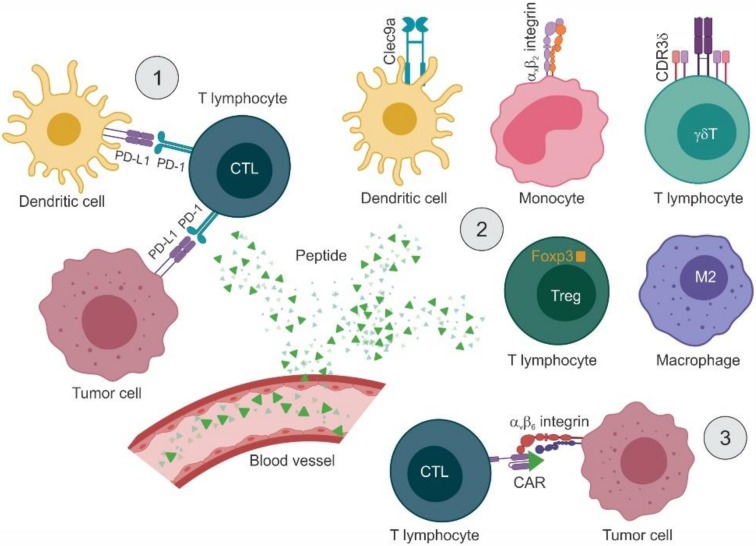
Phage display-derived peptides as nanomodulators of the immune response. Peptides administered intravenously can exit the blood vessels and reach different cell types to (**1**) inhibit PD-1/PD-L1 interaction or (**2**) target immune cells for either activation (dendritic cells, monocytes, λδT lymphocytes) or inhibition (Tregs, M2-like macrophages) of their functions. (**3**) Selected peptides can also be exploited in a CAR strategy. Abbreviations: PD-1, programmed cell death 1; PD-L1, programmed cell death ligand 1; CTL, cytotoxic T lymphocytes; CDR, complementary determining region; Tregs, T regulatory cells; CAR, Chimeric Antigen Receptor.

**Table 1 molecules-25-00843-t001:** Mimotopes of tumor-associated antigens.

Target	Ab	Peptide Motif	Reference
CD20	Rituximab	ITPWPHWLERSS	[29]
		QDKLTQWPKWLE	[30]
		WPxWLE (A/S)NPS	[31]
		WAANPS PYANPSL	[33]
		WPKWLE PYANPSL	[34]
EGFR	Cetuximab	CQFDLSTRRLKC CQYNLSSRALKC	[35]
	Cetuximab, Matuzumab	KTL YPLG	[37]
	12H23, Ch806	WHTEILKSYPHE LPAFFVTNQTQD	[38]
	ICR-62	QHYNIVNTQSRV	[39]
	Panitumumab	DTDWVRMRDSAR VPGWSQAFMALA	[41]
ErbB2	Trastuzumab	CQWMAPQWGPDC	[42]
		WxxGxAxGS	[43]
	L26, N12, L288	ALVRYKDPLFVWGFL	[44]
	SER4	INNEYVESPLYM	[45]
PSA	4G5	VDPGKYNKY EGPAKGFKL GCYEAPSKAAKC	[46]
		RRSHPCRTCTTHTP HRKTTCTRCPATSP HRRGECRACPLLPA RRPAHCHHCPRNP	[47]
CEA	Col-1	DRGGLWKTP	[17]
VEGF	Bevacizumab	DHTLYTPYHTHP	[48]
Gastric antigen MGb1	MGb1	HxQ LxS	[49]
Gastric antigen MG7	MG7	NAIYARNAQ TCHLRVYAQ SWAPVYARN	[50]
		KPHVHTK	[51]
		KPHLHFH KPHSHLH SWAPVYARAN	[53]
Human fibrosarcoma cell line	BCD-9	GRRPGGWWMR	[54]
Human B lymphoblastoid cell line	BAT	PRRIKPRKIMLG QKILQQINLPRI	[28,55,56]
Lc4Cer, nLc4Cer	AD117m, H11	RNVPPTFNDVYWIAF	[58]
GD1α	KA17	WHWRHRIPLQLAAGR	[59]
GD2	14G2a	EDPSHSLGLDVALFM	[60,61]
		NCDLLTGPMLCV SCQSTRMDPNCW VCNPLTGALLCS RCNPNMEPPRCF GCDALSGHLLCS	[64]
	ch14.18	CDGGWLSKGSWC CGRLKMVPDLEC	[65,66,67]
	ME361	LDVVLAWRDGLSGAS GVVWRYTAPVHLGDG	[68]
GD3	4F6	LAPPRPRSELVFLSV PHFDSLLYPCELLGC GLAPPDYAERFFLLS RHAYRSMAEWGFLYS	[69]
TF-Ag	JAA-F11	HIHGWKSPLSSL HHSHKTNLATTP GHPHYITHKPNR YPSLPVYHSLRS MHKPWSGHMQVP	[70]
Tumor-specific CD43 glycoforms	UN1	TCKLLDECVPLW SFAATPHTCKLLDEC-VPLWPAEG	[71]

**Table 2 molecules-25-00843-t002:** Phages as nanocarriers for anticancer vaccines.

Antigen	Epitope	Phage	Peptide Motif	Model	Reference
MAGE	MAGE-A1_161–169_	pfd8wf (*fd* derivative)	EADPTGFSY	B16-F10 melanoma	[85]
	MAGE-A10_254–262_	fd23 (*fd* derivative)	GLYDGMEHL	EL-4-HHD lymphoma	[87]
	MAGE-A3_271–279_	fd23 (*fd* derivative)	FLWGPRALV	EL-4-HHD lymphoma	[87]
	MAGE-1	T7	Entire antigen	4T1 breast carcinoma	[88]
P1A	P1A_35–43_	pc89 (*fd* derivative)	LPYLGWLVF	P815 mastocytoma	[89]
HER2	HER2_369-377_	λ	KIFGSLAFL	TUBO breast carcinoma	[90]
	HER2_776-790_	λF7	GVGSPYVSRLLGICL	TUBO breast carcinoma	[91]
MUC1	MUC1	Qß	PDTRPAPGSTAPPAHGVTSA	Non tumoral	[101]
VEGFR2	VEGFR2	T4	Extracellular portion	Lewis lung carcinoma	[103]
EGFR	EGFR	M13-pAK8-VIII (*fd* derivative)	QHYNIVNTQSRV	Lewis lung carcinoma	[104]
Cell line	Unknown	M13KE (*fd* derivative)	TRTKLPRLHLQS	B16/A2Kb melanoma	[79]
GnRH	GnRH	f8-8 (*fd* derivative)	EHPSYGLA	Non-tumoral	[105]
Tumor idiotype	B-cell receptor	M13K07 (*fd* derivative)	Entire antigen	BCL1 lymphoma	[108]
	Different myeloma idiotypes	M13K07 (*fd* derivative)	Entire antigen	Clinical trial in patients	[107]

**Table 3 molecules-25-00843-t003:** Phage display-derived peptides as direct nanomodulators of the immune response.

Target	Effect	Peptide Motif	Reference
IgM λ receptor	Activation of B-lymphoma cells	KP--xRV -W--WxR Y--EDLRRR --PVDHGL	[114,115]
Melanin	Reversal of melanin-induced immune response	HTTHHRN TTHQFPF NPNWGPR	[116]
PD-L1 and PD-1	Competitive inhibition of PD-1/PD-L1 binding	NYSKPTDRQYHFKHAHHTHNLRLP	[117]
		SGQYASYHCWCWRDPGRSGGSK	[118]
		VNYMSNQTKAPPGLSAILPYIQIE	[119]
		CLP002	[120]
		Affimer^®^ antagonists	[121]
		r_G9 repeabdy	[122]
CD11c/CD18 integrin α_X_β_2_	Targeting of APCs and induction of CTLs	CGRWSGWPADLC	[123,124]
Clec9a		WPRFHSSVFHTH	[125]
α_v_β_6_ integrin	CAR	RTDLDSLRTYTL	[127]
CDR3δ	Activation of γδT cells	WPHNWWPHFKVK PLLPMHPMKVSH KPPTQKRRRQTM RPRTRLHTHRNR YPWHWWHSVSPW FHWSWYTPSRPS WHHPWWYPRPGV	[128]
M2 TAMs	Reduction of infiltration by M2 TAMs	YEQDPWGVKWWY	[130]
Foxp3	Inhibition of Treg activity	RDFQSFRKMWPFFAM	[131,132]

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
