# Peer review of "Phage Display-Based Nanotechnology Applications in Cancer Immunotherapy"

_molecules, 2020, doi:10.3390/molecules25040843_

Round 1

Reviewer 1 Report

This is a very tranlational review in which the authors focused on  approaches based on peptide phage technology, performed with many variations from the original protocol and in combination with advanced computational modelling.

However the dimension of the nanotechnology is not clear. It is a new tool or an application?

Why  is Phage display a  nanotechnology science? A definition on peptide phage display nanotechnology in the first part is needed to introduce the subjet.

The conclusions of each section summarize the topic described in the parragraph and introduce the vision of the authors. A more in depth discussion is also needed to increase the added value of this work.

Globally it is a nice and complete work, but the nanotechnological dimension need to be detailed as well as the scope of this review.

Author Response

Response to Reviewer 1 Comments

Point 1: This is a very translational review in which the authors focused on approaches based on peptide phage technology, performed with many variations from the original protocol and in combination with advanced computational modelling. However the dimension of the nanotechnology is not clear. It is a new tool or an application? Why is Phage display a nanotechnology science? A definition on peptide phage display nanotechnology in the first part is needed to introduce the subject. […] Globally it is a nice and complete work, but the nanotechnological dimension needs to be detailed as well as the scope of this review.

Response 1: we thank Reviewer 1 for the constructive comments to our manuscript and for considering it “a nice and complete work”. Following his/her indications, in the updated version we now introduce a definition of phage display and discuss why this method is to be considered a nanotechnology (lines 71-82). We also better explain that, although representing a well-established tool, phage display can have different applications, among which the ones related to the new field of immuno-oncology described in the review. We also better define the scope of the review in a dedicated paragraph (lines 82-87).

Point 2: The conclusions of each section summarize the topic described in the paragraph and introduce the vision of the authors. A more in depth discussion is also needed to increase the added value of this work.

Response 2: in the updated version of the manuscript, we rearrange the conclusions for each paragraph, and we conclude with a partially rewritten final paragraph named “Concluding remarks”. To facilitate the reader through the different applications and molecular/cellular contexts, we add a table (Table 3) and tree new figures to replace and extend the previous Figure 1.

Reviewer 2 Report

This reveiw is well written and gives a lot of useful information about peptides and nanomedicines derived from phage display. It could be accepted as it is, but it might be better if authors can provide table of content, more example figures, and some summary tables of phage-based nanomedicine for readers.

Author Response

Response to Reviewer 2 Comments

Point 1: This review is well written and gives a lot of useful information about peptides and nanomedicines derived from phage display. It could be accepted as it is, but it might be better if authors can provide table of content, more example figures, and some summary tables of phage-based nanomedicine for readers.

Response 1: we thank Reviewer 2 for the very positive consideration of our work. As he/she suggested, the revised version of the manuscript now includes a total of three tables (one for each section) and three figures. As for the table of contents, to our understanding this should not be included in the manuscript; however, we will be glad to include also this part if required.